# Eicosapentaenoic Acid Regulates Inflammatory Pathways through Modulation of Transcripts and miRNA in Adipose Tissue of Obese Mice

**DOI:** 10.3390/biom10091292

**Published:** 2020-09-07

**Authors:** Theresa Ramalho, Mandana Pahlavani, Nishan Kalupahana, Nadeeja Wijayatunga, Latha Ramalingam, Sonia Jancar, Naima Moustaid-Moussa

**Affiliations:** 1Department of Immunology, Institute of Biomedical Sciences, University of São Paulo, São Paulo 05508-900, Brazil; ramalho.trr@usp.br; 2Department of Nutritional Sciences, Texas Tech University, Lubbock, TX 79409, USA; mandana.pahlavani@ttu.edu (M.P.); nishan.kalupahana@gmail.com (N.K.); nadeejaw@olemiss.edu (N.W.); lramalin@syr.edu (L.R.); 3Obesity Research Institute, Texas Tech University, Lubbock, TX 79409, USA; 4Department of Physiology, Faculty of Medicine, University of Peradeniya, Peradeniya 20400, Sri Lanka

**Keywords:** obesity, adipose tissue, inflammation, leukotriene-B4, eicosapentaenoic acid

## Abstract

This study aims to investigate the global profiling of genes and miRNAs expression to explore the regulatory effects of eicosapentaenoic acid (EPA) in visceral adipose tissue (VAT) of obese mice. We used male mice, fed either a high-fat diet (HF) or HF supplemented with EPA (HF-EPA), for 11 weeks. RNA, and small RNA profiling, were performed by RNAseq analysis. We conducted analyses using Ingenuity Pathway Analysis software (IPA^®^) and validated candidate genes and miRNAs related to lipid mediators and inflammatory pathways using qRT-PCR. We identified 153 genes differentially downregulated, and 62 microRNAs differentially expressed in VAT from HF-EPA compared to HF. Genes with a positive association with inflammation, chemotaxis, insulin resistance, and inflammatory cell death, such as *Irf5*, *Alox5ap*, *Tlrs*, *Cd84*, *Ccr5*, *Ccl9*, and *Casp1*, were downregulated by EPA. Moreover, EPA significantly reduced LTB4 levels, a lipid mediator with a central role in inflammation and insulin resistance in obesity. The pathways and mRNA/microRNA interactions identified in our study corroborated with data validated for inflammatory genes and miRNAs. Together, our results identified key VAT inflammatory targets and pathways, which are regulated by EPA. These targets merit further investigation to better understand the protective mechanisms of EPA in obesity-associated inflammation.

## 1. Introduction

Obesity is a complex chronic disorder with a multifactorial etiology, including genetics, age, hormones, diet, and environment. It is a major public health challenge globally, affecting approximately 40% of adults in the United States of America [1,2]. In this condition, hypertrophic adipose tissue stores excess lipids and become inflamed with dysregulated levels of inflammatory mediators. Specifically, there is a reduction of anti-inflammatory markers production, such as interleukin (IL)-10 and adiponectin, with increased production of pro-inflammatory markers, such as tumor necrosis factor-alpha (TNF-α), IL-6, and monocyte chemoattractant protein (MCP-1) [3].

Dietary bioactive compounds, such as fish oil, contain long-chain ω-3 polyunsaturated fatty acids (PUFAs) with anti-inflammatory properties. Among these PUFAs, eicosapentaenoic acid (20:5n-3, EPA) is a ω-3-PUFA with anti-inflammatory, cardioprotective, and triglyceride-lowering properties [4]. Omega 3-PUFAs metabolites, such as E-resolvins, derived from EPA metabolism, along with D-protectins and maresins, derived from docosahexaenoic acid (DHA), also have inflammation-resolving properties [4]. We have previously reported that enrichment of high fat (HF) diet with EPA reduced body weight, adiposity, glucose intolerance, and inflammation, while improving insulin sensitivity in obese mice to levels comparable to low-fat fed mice [5,6]. In these studies, we showed that EPA supplementation incorporated EPA, DHA, docosapentanoic acid (DPA) and prostaglandin (PG)F1α in adipose tissue of HF fed mice [6].

An HF diet differentially regulates genes and miRNA in different adipose depots. Such regulation promotes inflammatory responses, endoplasmic reticulum stress, lipogenesis, and adipogenesis in visceral adipose tissue (VAT) [7]. However, to our knowledge, exploration of the regulatory effects of EPA in VAT in obesity using global profiling of gene and miRNA expression has not been reported. We hypothesized that EPA regulates novel metabolic and inflammatory pathways through these miRNAs and target genes, mediating protective metabolic and anti-inflammatory effects in obesity. We performed a global profiling of genes and miRNA from the VAT of mice fed an HF diet and HF diet supplemented with EPA in an attempt to find possible correlations of gene profiling and miRNAs with inflammatory/metabolic changes caused by EPA in VAT.

## 2. Materials and Methods

### 2.1. Animals and Experimental Design

Male C57BL/6 mice (Jackson Laboratory, Bar Harbor, ME, USA) aged 5–6 weeks were fed an HF (45% kcal from fat, 35% kcal from carbohydrate, 20% kcal from protein) or an HF diet supplemented with 36 g EPA/kg diet (6.57% kcal; HF-EPA) (Research Diets, New Brunswick, NJ, USA) for 11 weeks. Various methods were used to translate mouse dietary intakes into human equivalent intakes, as we detailed in the discussion section. Detailed diet compositions are available in Kalupahana et al., 2010 [5]. After 11 weeks, mice were sacrificed for the epididymal fat pad, referred to as VAT, isolated, and stored at −80 °C for further analyses. These protocols were approved by the Institutional Animal Care and Use Committee at the University of Tennessee, Knoxville (TN), and Texas Tech University, Lubbock (TX).

### 2.2. RNA Extraction and cDNA Library Preparation

RNA was isolated from VAT using RNeasy Mini Kit (QIAGEN, Redwood City, CA, USA) and quantified using Nanodrop, (Thermo Scientific, Waltham, MA, USA). Agilent 2200 Tape station (Agilent Technologies, Santa Clara, CA, USA) was used to assess RNA quality of VAT for RNA sequencing, and those with RNA integrity numbers (RIN) between 7.1 and 8.2 were used. RNA-Seq was performed using RNA from VAT and of HF and HF-EPA mice (*n* = 3 per group). 2–4 μg of RNA per sample was used to make cDNA libraries using Illumina’s TruSeq RNA sample preparation kit v2 (Illumina Inc., San Diego, CA, USA). cDNA was recovered using Agencourt, AMPure XP beads (Beckman Coulter, Indianapolis, IN, USA). Specific index was used for each sample, and then the adaptor-ligated fragments were purified using AMPure XP beads and amplified with polymerase chain reaction (PCR). Validation and insert size identification were performed for the cDNA libraries using the Agilent 2200 Tape station (Agilent Technologies, Santa Clara, CA, USA). Libraries were quantified using Qubit 2.0 fluorimeter (Invitrogen, Life Technologies, Waltham, MA, USA).

### 2.3. mRNA and miRNA Sequencing Strategy

Paired-end sequencing was performed using Illumina HiSeq 2500 (Illumina, San Diego, CA, USA) with a 108 bp read length for transcriptomic studies (Center for Biotechnology and Genomics Core Facility, Texas Tech University, Lubbock, TX, USA). Illumina small RNA protocol for miRNA profiling, (Illumina, Inc., San Diego, CA, USA) was used for libraries preparation and sequencing. Illumina Genome Analyzer NextSeq 500 (Illumina, Inc., San Diego, CA, USA) was used to perform miRNA sequencing (Department of Molecular and Cell Biology, Baylor College of Medicine, Houston, TX, USA).

### 2.4. Quality Control (QC) and Reads Alignment

FastQC high throughput sequence QC report (Version 0.11.2) (http://www.bioinformatics.babraham.ac.uk/projects/fastqc/) was used to check the quality of raw sequence reads for gene expression. The average base call quality Q score based on FastQC data was 38. The reads were mapped to the Mus musculus genome (GRCm38.p4) using Qseq^®^ software Version 12 (DNASTAR, Madison, WI, USA). The Gunaratne Next Generation pipeline was used to identify miRNA expression [8]. For the quality control of Fastqc data for small RNA sequencing, counts of each unique read of miRNA were normalized to total usable reads, then 40 counts were added. Normalized expression values of genes and miRNAs were presented as reads per kilo base per million mapped reads (RPKM).

### 2.5. Functional Analysis

We entered the RPKM values, fold change, and *p* value obtained from a double-blind analyzer into an Excel filter tool to identify differentially expressed genes from the global transcriptome. We used the filters “show numbers >0.05”, “show numbers >0.3”, and “show numbers >1.5 or show numbers <0.6” to identify differentially expressed genes, confident expression, and minimum fold change, respectively. For miRNAs, the filters were “show numbers >0.05”, “show numbers >3.34” and “show numbers >1.25” to identify differentially expressed miRNAs, confident expression, and minimum fold change, respectively. TIGR Multiple Experiment Viewer (MeV) v4.8.1 was also performed to identify DE genes and miRNAs using Hierarchical clustering [9].

### 2.6. Data Availability

RNA-seq data for HF and HF-EPA were submitted in the NCBI Sequence Read Archive (SRA) under BioProject (NCBI, PRJNA350826). Individual accession numbers for HF Sequence Read Archive (SRA) submission: VAT1 (SAMN05717648), VAT2 (SAMN05717649), and VAT3 (SAMN05717650). For HF-EPA SRA submission: VAT-EPA1 (SAMN05991735), VAT-EPA2 (SAMN05991736), and VAT-EPA3 (SAMN05991737) (http://www.ncbi.nlm.nih.gov/bioproject/). Small RNA data for miRNA profiling has been submitted to the Gene Expression Omnibus (GEO, accession number GSE85101) (http://www.ncbi.nlm.nih.gov/projects/geo/).

### 2.7. Quantitative Real-Time Polymerase Chain Reaction (q-PCR) Validation

RNA was isolated from VAT followed by cDNA synthesis using RNeasy kit (QIAGEN, Redwood City, CA, USA) using the iScript kit (Bio-Rad, Inc., Irvine, CA, USA) and TaqMan^®^ Advanced miRNA cDNA Synthesis Kit (Life Technologies Corporation, Pleasanton, CA, USA) for genes and miRNAs, respectively. Bio-Rad-CFX96 real-time PCR (Bio-Rad, Inc., Irvine, CA, USA) was used to perform qPCR. For genes and microRNAs, 18S, and miR-191-5p were used as housekeeping control, respectively.

### 2.8. Enzymatic Assay for LTB4 Dosage

VAT was processed in RIPA solution (Cell Signaling, Danvers, MA, USA), and ethanol (2:1) was used to precipitate proteins. Ethanol was then evaporated under nitrogen, and LTB4 was measured using LTB4 EIA Kit (Cayman, Ann Arbor, MI, USA) following the manufacturer’s instruction.

### 2.9. Statistical Analyses

Moderated t-test with false discovery rate (FDR) of 5% (using Benjamini—Hochberg test) and QSeq software were used to identify differentially expressed (DE) genes between HF (control) and HF-EPA (1.5 or more-fold change at 95% confidence; and RPKM ≥ 0.3 were used in both HF and HF-EPA as the threshold for gene expression levels). Based on Student’s *t*-test, *p* ≤ 0.05 was considered significantly different when qPCR gene expression was compared between HF and HF-EPA groups. For analysis of pathways, genes, and networks, the Ingenuity Pathways Analysis (IPA^®^, QIAGEN Redwood City, CA, USA) software was used. IPA was used to identify nucleotide sequence- and literature-based networks among transcripts and identify nucleotide sequence- and literature-based networks between miRNAs and mRNAs. Fischer’s exact test *p* values indicate the significance of enrichment of pathways by DE genes, in the canonical pathway analyses. Based on the DE genes in the dataset, Z score ≤ −2 indicates that the canonical pathway is inhibited, while Z ≥ +2 is activated. A cut off of log2 fold change > 1.25 was used for miRNAs to determine DE ones between HF vs. HF-EPA, RPKM ≥ 3.34. Differentially expressed miRNAs were submitted to canonical pathways using the grow function in IPA^®^, and the miRNA model was based on bioinformatic analysis of its binding sequences in the genes discussed in this study. For the q-PCR, results are presented as means ± SEM and the ∆∆Ct method to determine mRNA expression and fold changes. Three to five replicates were used for each dietary or treatment groups.

## 3. Results

### 3.1. RNA Sequencing Data Quality

VAT samples used in this research were from our previously conducted study [5]. Total sequences per sample mapped were about 24 million for RNA sequencing. A total of 27151 transcripts were identified in VAT (HF and HF-EPA). For small RNA sequencing, an average of 13.8 million sequences reads per sample was mapped to the *Mus Musculus* genome (build mm 10), and a total of 742 miRNAs were identified in VAT (HF and HF-EPA).

### 3.2. Effects of EPA on Gene and miRNA Expression of VAT of HF Fed Mice

HF and HF-EPA groups expressed a total of 13,900 and 13,251 genes, respectively (Figure 1A). Out of all the 27,151 genes identified in the transcriptome, 153 were DE between HF and HF-EPA, with all of them downregulated with EPA compared to HF (Appendix A, Figure 1A). Parts of the DE genes are shown in Table 1, Table 2 and Table 3. Four genes were upregulated by EPA, but three of them had RPKM value below 0.3 and were not included in Tables with DE genes. The genes with RPKM values below 0.3 were included in Appendix A, which also includes only one gene upregulated by EPA with RPKM value above 0.3 in the HF-EPA group. In the miRNA analysis, we found a total of 742 miRNAs expressed in both groups, out of which, 379 and 363 miRNAs were exclusively expressed in HF and HF-EPA, respectively (Figure 1B). We identified 62 DE miRNAs with a log2 fold change >1.25 (*p* ≤ 0.05). Out of the 62 DE miRNAs, 31 were upregulated (Appendix A), and 31 were downregulated with EPA (Appendix A), respectively, compared to HF (Figure 1B). Based on the DE of genes and miRNAs, we performed hierarchical clustering (HC) to determine the similarity in expression level in different groups (HF and HF-EPA), which is represented in different colors (Figure 1A,B).

### 3.3. The Effects of EPA on Canonical Pathways and Their Genes

According to the analysis of canonical pathways affected by EPA supplementation in an HF diet, EPA altered seven pathways in VAT of the HF-diet-fed mice, as shown in Table 1. Part of the analysis of the transcripts in this study was based on the data in Table 1. According to the data, EPA downregulated all the genes shown in canonical pathways analysis. We presented RPKM, *P* values, and fold change of such genes in Table 2 for surface receptors, and Table 3 for intracellular proteins. Among the pathways, the “Fcγ Receptor-mediated Phagocytosis in Macrophages and Monocytes” appeared negatively affected. “Triggering Receptor Expressed on Myeloid Cells (TREM1) signaling” were differentially downregulated by EPA. In general, TREM activation culminates in Phosphoinositide 3-Kinase (PI3K) and Nuclear Factor Kappa B (NFkB) signaling, both classically characterized in immune cells and inflammation [10]. Interestingly, PI3K and NFkB appeared in our canonical pathways analysis for pathways affected by EPA supplementation as “PI3K Signaling in B Lymphocytes” and “NFkB signaling”. Similarly, “CD28 Signaling in T Helper Cells” and “Role of Nuclear factor of activated T-cells (NFAT) in Regulation of the Immune Response” were also downregulated, indicating that EPA may exert direct or indirect effects on lymphocytes, in addition to monocytes/macrophages cells. Interestingly, EPA also negatively regulated “Neuroinflammation Signaling Pathway”, and some of the genes that showed up in this pathway, are involved in inflammasome activation, such as *NLR family apoptosis inhibitory protein* (*Naip2*) *and PYD and CARD Domain Containing* (*Pycard*) (Table 1 and Table 3).

All the pathways downregulated by EPA are involved in inflammatory processes. Some genes appeared redundant among the pathways, such as *Pik3*-associated genes (*Phosphoinositide-3-Kinase Regulatory Subunit 5—Pik3r5* and *Phosphoinositide-3-Kinase Adaptor Protein 1*—*Pik3ap1*), which appeared in five of the seven pathways (Table 1 and Table 3); inflammasome-related genes (*Caspase 1*—*Casp1*, *Caspase Recruitment Domain Family Member 11*—*Card11*, *Naip2,* and *Pycard*) appeared in four pathways (Table 1 and Table 3); *Toll like receptors* (*Tlr1*, *Tlr6*, *Tlr7*, *Tlr8,* and *Tlr13*) showed up in three pathways (Table 1 and Table 2); and *Fc Fragment of IgG Receptor*-related genes (*Fcgr1a* and *Fcgr3a/b*) were identified in two pathways (Table 1 and Table 2).

### 3.4. Further Genes and Networks

In addition to the genes that appeared in the canonical pathways analysis, we identified further DE genes downregulated by EPA (Table 2 and Table 3). Among the genes, we highlighted the *Irf5*, which is negatively correlated with insulin sensitivity and obesity [11], and *Alox5ap*, which signals adipose tissue inflammation and lipid dysfunction in adipose tissue of obese mice [12]. Another important receptor downregulated by EPA is *Cd84*, a cell-surface molecule that activates cytokine production by macrophages through signal transducer and activator of transcription 1 (STAT1) activation. Chemokines are also fundamental players in the recruitment of inflammatory cells in hypertrophic adipose tissue [13]. EPA downregulated *Ccr5* and *C-C motif chemokine 9* (*Ccl9*). The RPKM, *P* values, and fold change of *Irf5* and *Alox5ap* were included in Table 2, and *Cd84* and *Ccr5* were included in Table 3.

The genes mentioned in this and the previous section compose an important network that may contribute to the comprehension of the effects of EPA on the inflammatory process of the adipose tissue of HF fed mice. We previously reported lower expression of key obesogenic factors in VAT from HF-EPA mice compared to HF group, such as plasminogen activator inhibitor-1 (PAI-1), monocyte chemoattractant protein 1 (MCP-1) at protein levels [5], *Galectin-3 in macrophages* (*Lgals3*), *Bone morphogenic protein 4* (*Bmp4*), *CCAAT-enhancer binding protein alpha* (*Cebpa*), *Leptin* (*Lep*) and *Sterol Regulatory Element Binding Transcription Factor 1* (*Srebf1*) in mRNA levels [6]. Other studies have also shown the relevance of EPA for the modulation of pro-inflammatory lipid mediators synthesis [14]. In the validation with qPCR, we identified that, compared to HF, the HF-EPA group had significantly lower expression of genes positively correlated with adipose tissue hypertrophy and inflammation, such as *Arachidonate 5-lipoxygenase* (*Alox5*), *Myeloid differentiation protein 88* (*Myd88*), *Stat1*, *Ccr5*, and *Card11* (*p* ≤ 0.05, Figure 2A). We also observed lower production of LTB4 in adipose tissue from the HF-EPA group, compared to HF (Figure 2B), consistent with reduced *Alox5* expression.

### 3.5. Identification of microRNAs, Validation, and Networks

Based on our findings in which EPA exerts effects on inflammatory pathways and their markers, we focused our analysis on the microRNAs involved in inflammation (Table 4, Figure 3). Of these, miR-30a-5p and miR-143-3p shown in Table 4, were upregulated by EPA in the small-RNA seq data, and validated using qPCR (Figure 3). However, miR-221-3p, miR-125b-5p, miR- 146b-5p, and 181b-5p, shown in Table 4, were downregulated by EPA in the small-RNA seq data, and were validated using qPCR (Figure 3). To our knowledge, out of these, only miR-221 was reported in the literature to be regulated in response to EPA [15].

Further analysis of the DE miRNAs was conducted in the canonical pathway function in IPA to create networks between genes and miRNAs (Figure 4). These networks were also based on previous literature and genes identified in the previous section. It has been previously reported that inflammatory factors that culminate in inflammasome activation, such as *Tlrs* [16], leukotrienes [14], cytokines [17], and chemokines [13] are downregulated by EPA. Also, in our network analysis, miR-221, miR-125b-5p, and miR-146b were downregulated by EPA, consistent with its role in obesity and inflammation in hypertrophic adipose tissue [18,19,20]; while miR-30c-5p was upregulated by EPA, corroborating with its function in the improvement of inflammation in adipose tissue [21]. *Alox5* appeared in the networks shown in IPA and LTB4-related genes (*Myd88*, *Stat1,* and *NFkB*) also appeared as intermediates linking the pathways based on gene and miRNAs that we found. Based on this, we built the Figure 4 network, and evaluated LTB4 levels in VAT, gene expression (qPCR) of the main intermediaries, and key transcripts that compose such networks (Figure 2 and Figure 3).

## 4. Discussion

This study reports the impact of EPA supplementation in an HF-diet-fed mice by gene and miRNA profiling analysis in VAT. Gene and miRNA profiling performed in this study revealed pathways and networks associated with inflammation and adipose tissue hypertrophy regulated by EPA. Our findings suggest that EPA reduces expression of inflammatory markers involved in chemotaxis, inflammation, and cell death.

Our group has previously reported that EPA negatively regulates inflammatory mediators in mouse hypertrophic adipose tissue. In the metabolomic analysis by LC-MS/MS and GC-MS, we have shown that EPA supplementation changes epididymal adipose tissue free fatty acids content [6]. Such changes involve an increase in EPA, ω3-DPA, and acetylcarnitine, and reduction in ω6-DPA and PGF1α [6]. These data suggest that the alterations observed in transcripts and miRNAs in this study may be due to FFA incorporation promoted by EPA in the adipose tissue of HF fed mice. We have previously reported significant reductions in body weight, basal glycemia, insulinemia, insulin resistance (evaluated by homeostatic model assessment for insulin resistance score—HOMA-IR), glucose clearance, and body fat in mice fed HF-EPA compared to HF [5,6]. This indicates that the effects of EPA on inflammatory pathways may reflect the improvement in adiposity and insulin sensitivity in HF-EPA fed mice.

Considering the inflammatory pathways by which EPA may exert effects in the adipose tissue of HF mice, there is an appeal for EPA effects on immune cells, such as T and B lymphocytes, monocytes, and macrophages. There is evidence for the effects of EPA on immune cells [22]. We speculate that EPA would affect immune cells directly, or indirectly through alterations of lipid content in adipocytes. However, we have not explored single isolated immune cells to confirm. Several studies have shown the importance of macrophages in the inflammation of hypertrophic adipose tissue, due to the capability of these cells to secrete cytokines, chemokines, and growth factors [23]. There are several subtypes of macrophages in the hypertrophic fat. TREM2+ macrophage is a subtype that downregulates adipocyte growth [24]. Interestingly, *Trem2* expression was downregulated by EPA in our data. *Il1rn* (named Interleukin 1 Receptor Agonist (IL-1RA) when protein is expressed) is another regulatory marker in macrophages of hypertrophic adipose tissue [25]. Possibly, EPA downregulated *Il1rn* and *Trem2*, due to its effects on reducing tissue hypertrophy, when *Ilrn1* and *Trem2* are no longer needed for tissue auto-regulation. The same can be speculated for the TREM-1 pathway, however, no study has investigated how TREM1 affects adipose tissue in the condition of obesity.

Regarding the PI3K signaling in B lymphocytes, there is evidence for the role of active B cells in the milieu of inflamed fat [26]. In such evidence, LTB4 secreted by B cells exerts important effects on obese adipose tissue [26]. Our study shows that EPA reduced the expression of *Alox5ap*, a protein that activates LTB4 synthesis by activating 5-lipoxygenase (5-LO), and promotes adipose tissue inflammation and lipid dysfunction in obese mice [12]. Our validation data also showed that although *Ltb4r1* expression was not changed, *Alox5* expression was reduced by EPA. Moreover, EPA reduced components of the LTB4 pathway, such as MyD88, *Stat1,* and *Ccr5*. Accordingly, EPA reduced LTB4 levels in adipose tissue of HF mice. These data suggest that EPA may downregulate inflammatory pathways though negatively affecting the synthesis machinery of LTB4, since EPA and other free fatty acids (FFA) are upregulated by EPA supplementation in hypertrophic adipose tissue. Indeed, it is well documented that LTB4 synthesis is downregulated by EPA supplementation [27,28].

Regarding the effects of EPA on miRNAs expression, we found that EPA reduced the expression of miR-146b and miR-125b-5p in the adipose tissue of HF mice. A study showed that LTB4 promotes higher expression of miR-146b and 125b-5p [29]. Therefore, our data suggest that reduction of miR-146b and miR-125b-5p could probably be through the effects of EPA on the reduction of LTB4 levels. In our data, EPA upregulated miR-30a-3p, which indirectly downregulates inflammatory targets, such as c-jun and c-fos (part of MAPK signaling pathway) [30]. Although c-jun and c-fos were not detected in our transcriptome, we found that EPA downregulated its upstream activator, *Pik3ap1* [31]. This suggests that the increase in miR-30 promoted by EPA negatively affects Pi3kap1 expression. Another miRNA modulated by EPA is miR-221-3p. A recent paper showed that miR-221 is induced by c-fos and c-jun activation [32]. Since miR-30, which targets c-jun and c-fos, was indirectly upregulated by EPA, we suggest that EPA reduced expression of miR-221 indirectly in response to EPA effects on miR-30. Interestingly, both miR-30 and miR-221 are implicated in adipogenesis in obesity [18,33]; miR-30 is involved in protecting inflammation in obesity, while miR-221 is associated with inflammation in adipose tissue [18,33].

It is important to mention that all the pathways identified in this study, including the LTB4 pathway, involves genes linked directly or indirectly with NF-kB, which leads to the transcription of inflammasome/cell death markers [34]. We observed EPA effects on such markers. It is known that in inflamed adipose tissue, cell death is activated in hypertrophic adipocytes [35]. This leads to a phenomenon in which efferocytosis is activated in macrophages to engulf necrotic adipocytes and develop crown-like structures, characterized by cytosolic lipid droplets [35]. The reduced expression of cell death markers, such as *Naip2*, *Pycard*, *Casp1*, and also *Card11,* by EPA corroborates our previous data showing reduced crown-like structures in gonadal adipose tissue of HF-EPA fed mice [6]. The reduction of such markers is also in accordance with a recent study in which ω-3 fatty acids suppresses NLRP3 inflammasome signaling in adipose tissue of individuals with obesity [36]. It is also important to emphasize that LTB4 is also involved in inflammasome activation in a model of chronic inflammatory disease [37]; therefore, lower expression of cell death markers by EPA may be associated with the lower levels of LTB4 in adipose tissue.

There are a few limitations in our study, which include the lack of analysis of samples of female mice and the effects of EPA at the proteome level. Such limitations merit future investigations. Based on previous reports and on the main EPA targets found in this study, several cytokines and chemokines are affected systemically by EPA. Different studies have reported elevated systemic levels of IL-1RA that can be released by adipose tissue in individuals with obesity [38,39,40,41]. Since our transcriptome data showed EPA reducing *Il1rn* expression in VAT of HF mice consistent with previous reports [42], we speculate that EPA would reduce IL-1RA levels in the circulation of HF mice. Thus, it is likely that IL-18 is similarly changed. First, IL-18 is usually found in the circulation of individuals with obesity [43]. Second, we report that EPA reduced inflammasome markers in VAT, which would reduce IL-18 release from adipose tissue. Indeed, reduced systemic levels of IL-18 when EPA is added to the diet of subjects with obesity has been reported [42]. TNF-α is another cytokine whose elevated levels are downregulated by EPA [42]. Our transcriptome and miRNA data have shown EPA affecting NFkB signaling, which drives TNFα transcription [34]. Therefore, TNFα systemic levels would also be reduced by EPA indirectly through NFkB regulation in our obesity model. The same could be suggested for MCP-1, since our lab and others have previous reports of EPA reduced MCP-1 along with macrophage content in adipose tissue and serum of diet-induced obese mice [5,44]. CCL5 is another chemokine suggested as a target for EPA in our model. CCL5 transcription is also induced by NFkB. In agreement with this, EPA reduced the expression of CCL5 receptor, *Ccr5*, and other markers of NFkB pathways in our data.

Another limitation of our study is the high dose of EPA used. Previously [5,6], we estimated this dose based on the caloric intake of fat as follows—36 g EPA /kg mouse diet represents 6.7% of total energy for mice fed 45% kcal fat diets. If we consider the upper DRI intakes of 35% kcal fat diets for human intake, the amount of EPA would be ~5% kcal EPA (total energy), or 100 kcal based on a 2000 kcal daily energy intake, or ~10–11 g EPA/day of EPA. However, other methods have been recommended in recent years, for translating mouse doses to humans. One of these methods is based on the body surface and body weights for mice (or other animal models) compared to humans [45,46]. If we consider this method, our dose of 36 g EPA/kg diet (for a 30 g mouse and 2.5 g food intake per day) is estimated to be 14g EPA/day for a 60 kg human body weight).

In summary, our study is the first report of integrated mRNA-miRNA analyses of EPA effects in diet-induced obese mice. While additional validations and functional assays are needed, this is the first data-driven study that revealed targets and pathways which integrate anti-inflammatory targets of EPA on obesity. Such findings contribute to new knowledge about mechanisms of action of omega-3 PUFAs in obesity and identified targets could provide new venues for anti-obesity therapies.

## 5. Conclusions

This study addressed EPA regulatory effects on several previously known genes, along with a few new genes, miRNAs, and pathways involved in inflammation—which lead to dysregulated metabolic homeostasis. We used a novel approach for examining the impact of EPA supplementation in an HF-diet-fed mice, by performing both transcriptomic and miRNA profiling. In addition to the analysis of differentially expressed genes and miRNAs, we conducted pathway analysis that provides a comprehensive understanding of metabolic and molecular changes mediated by EPA. Specifically, we identified inflammation-related targets in VAT that pave the way for further mechanistic studies for the potential use of EPA in obesity associated-inflammation and metabolic dysfunctions.

## Figures and Tables

**Figure 1 biomolecules-10-01292-f001:**
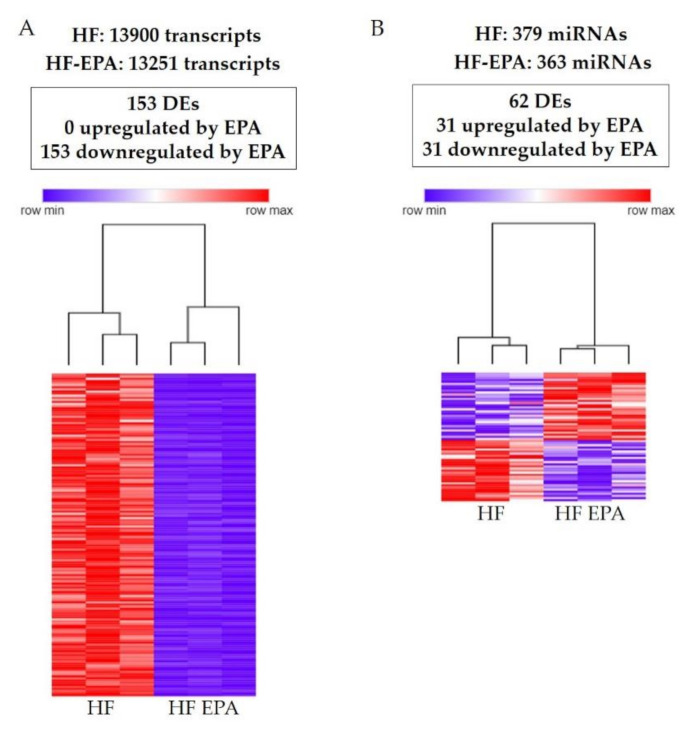
Gene and miRNA profiling and Hierarchical clustering process in obese B6 mice. (**A**). 153 DE genes identified following RNA sequencing for HF vs. HF-EPA in visceral adipose tissue (VAT). Zero genes were upregulated and 153 were respectively downregulated with EPA (false discovery rate (FDR)-adjusted *p* value ≤ 0.05, fold change ≥ 1.5, reads per kilo base per million mapped reads (RPKM) ≥ 0.3). Hierarchical clustering shows that 153 DE genes in VAT were differentially regulated by EPA. The horizontal line represents the different diets (HF vs. HF-EPA), and the vertical line represents the list of genes. Lower values are represented with blue tones, while higher values have yellow tones (FDR-adjusted *p* value ≤ 0.05, fold change ≥ 1.5, RPKM ≥ 0.3). (**B**) 62 DE miRNAs were identified following RNA sequencing for HF vs. HF-EPA in VAT. Thirty-one miRNAs were upregulated and 31 were respectively downregulated by EPA (FDR-adjusted *p* value ≤ 0.05, log2 fold change > 1.25, RPKM ≥ 3.34). Hierarchical clustering demonstrates that 62 DE miRNAs in VAT were differentially regulated by EPA. The horizontal line represents the different diets (HF vs. HF-EPA), and the vertical line represents the list of miRNAs. Lower values are represented with blue tones, while higher values have yellow tones (FDR-adjusted *p* value ≤ 0.05, log fold change > 1.25, RPKM ≥ 3.34). HF, high fat; HF-EPA, HF diet supplemented with EPA; DE, differentially expressed.

**Figure 2 biomolecules-10-01292-f002:**
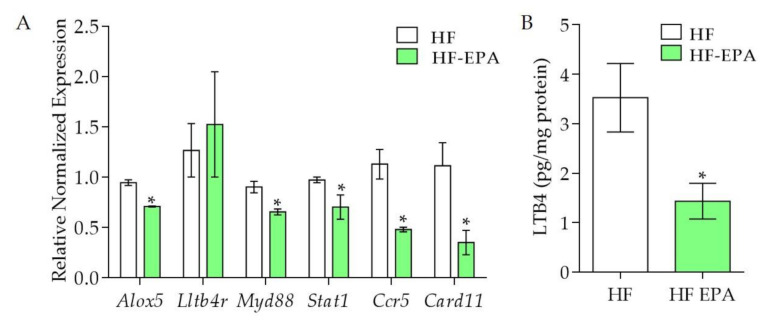
Validation of genes and LTB4 levels in adipose tissue from obese B6 mice. The expression of genes was validated by Q-PCR. (**A**) Gene expression of inflammatory markers *Alox5*, *Ltb4r*, *Myd88*, *Stat1*, *Ccr5*, and *Card11*; data are expressed as mean ± SEM, *p* ≤ 0.05, n = 6. (**B**) LTB4 levels in adipose tissue. Data are expressed as mean ± SEM, * *p* ≤ 0.05, n = 6; HF vs HF-EPA.

**Figure 3 biomolecules-10-01292-f003:**
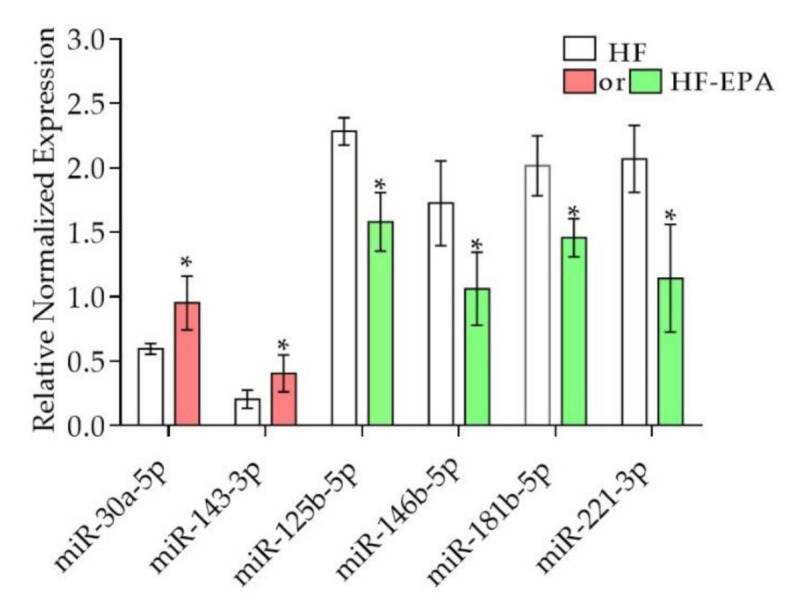
Validation miRNAs in adipose tissue of obese B6 mice using qPCR. The miRNA expression levels were validated. Validation of miR-30a-5p, miR-143-3p, miR-125b-5p, miR-146b-5p, miR-181b-5p, miR-221-3p; data are expressed as mean ± SEM, * *p* ≤ 0.05, n = 6; HF vs. HF-EPA. Red = upregulated; green = downregulated.

**Figure 4 biomolecules-10-01292-f004:**
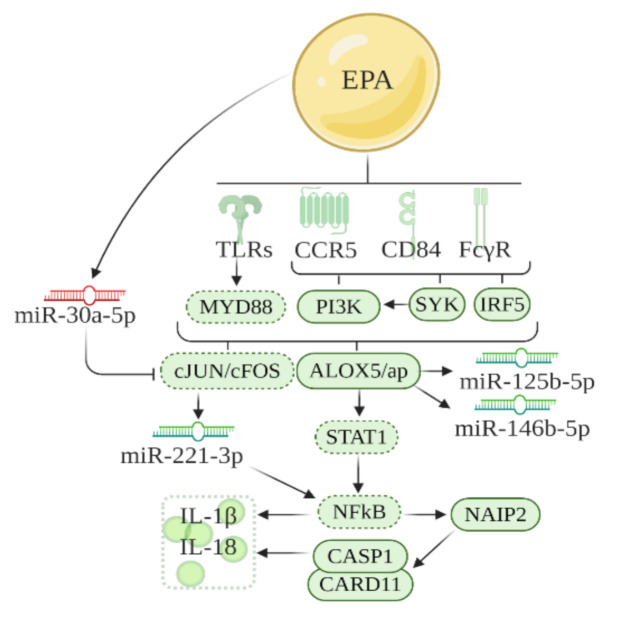
Schematic representation of the pathway regulated by EPA through modulation of genes and miRNAs in inflammatory pathways in VAT of obese B6 mice. The network is designed based on the IPA analysis and past literature. Solid lines around boxes, designed receptors, and miRNAs represent genes and miRNAs identified by our study, while dashed lines, represent genes from published literature. Red = upregulated; green = downregulated.

**Table 1 biomolecules-10-01292-t001:** Canonical pathways affected by EPA supplementation in visceral adipose tissue of obese B6 mice. −log *p* value > 1.3 is equivalent to FDR-adjusted *p* value ≤ 0.05, Z score ≤ −2 indicates pathway inhibited, Ratio is the number of differentially expressed/total number of genes in a canonical pathway.

Ingenuity Canonical Pathway	Gene Name	−log (*p*-Value)	Ratio	z-Score
**Fcγ Receptor-mediated Phagocytosis in**	*Fcgr1a, Hck, Fcgr3a/b, Ncf1, Syk, Pak1*	5.520	0.05	−2646
**Macrophages and Monocytes TREM1 Signaling**	*Tlr8, Tlr1, Tlr13, Lat2, Tlr7, Tlr6, Casp1*	6.190	0.0933	−2646
**CD28 Signaling in T Helper Cells**	*Card11, Syk, Pik3r5, Pak1*	2.700	0.0373	−2
**PI3K Signaling in B Lymphocytes**	*Atf3, Cd180, Pik3ap1, Blnk, Syk*	2.650	0.0362	−2236
**Neuroinflammation Signaling Pathway**	*Il1rn, Tlr8, Cd200r1, Tlr1, Trem2, Csf1r, Tlr13, Tlr7, Pik3r5, Tlr6, Naip2, Pycard*	7.660	0.0447	−3051
**Role of NFAT in Regulation of the Immune Response**	*Fcgr1a, Blnk, Fcgr3a/b, Syk, Pik3r5*	3.480	0.0357	−2646
**NF-κB Signaling**	*Il1rn, Card11, Tlr8, Tlr1, Tnfrsf11a, Tlr7, Pik3r5, Tlr6*	4.400	0.0421	−2828

Abbreviations: Fc Fragment of IgG Receptor (Fcgr), Hemopoietic Cell Kinase (Hck), Neutrophil Cytosolic Factor (Ncf), Spleen Associated Tyrosine Kinase (Syk), P21 (RAC1) Activated Kinase 1 (Pak1), Toll like receptors (Tlr), Linker For Activation of T Cells Family Member 2 (Lat2), Caspase 1 (Casp1), Caspase Recruitment Domain Family Member 11 (Card11), Phosphoinositide-3-Kinase Adaptor Protein 1 (Pik3ap1), Phosphoinositide-3-Kinase Regulatory Subunit 5 (Pik3r5), Activating Transcription Factor 3 (Atf3), Cd180 molecule (Cd180), B Cell Linker (Blnk), Cell Surface Glycoprotein CD200 Receptor 1 (Cd200r1), Triggering Receptor Expressed On Myeloid Cells 2 (Trem2), Colony Stimulating Factor 1 Receptor (Csf1r), NLR family apoptosis inhibitory protein (Naip2), PYD and CARD Domain Containing (Pycard), Interleukin 1 Receptor Antagonist (Il1rn), Tumor Necrosis Factor Receptor Superfamily, Member 11a, NFKB Activator (Tnfrsf11a).

**Table 2 biomolecules-10-01292-t002:** Differentially expressed genes data of surface receptors downregulated by EPA in visceral adipose tissue of B6 obese mice. FDR-adjusted *p* value ≤ 0.05, fold change ≥ 1.5 at 95% confidence.

Gene Name	HF RPKM	HF-EPA RPKM	Fold Change	*p* Value
*Fcgr1a*	27.683	3.811	0.137	0.0393
*Fcgr3a/b*	148.474	36.322	0.244	0.0449
*Tlr1*	13.167	2.279	0.17	0.0351
*Tlr6*	4.133	1.191	0.28	0.0460
*Tlr7*	5.472	1.44	0.26	0.0385
*Tlr8*	10.442	1.384	0.13	0.0296
*Tlr13*	10.023	2.192	0.21	0.0377
*Il1rn*	30.568	1.996	0.065	0.0248
*Csf1r*	139.035	28.388	0.20	0.0293
*Tnfrsf11a*	6.123	1.284	0.20	0.0385
*Cd200r1*	21.775	3.285	0.15	0.0173
*Cd180*	27.396	4.421	0.161	0.0218
*Cd84*	78.71	18.594	0.23	0.0373
*Ccr5*	22.7	3.087	0.13	0.0421
*Trem2*	254.804	46.296	0.18	0.0460
*Lat2*	69.723	15.287	0.21	0.0439

Abbreviations: Fc Fragment of IgG Receptor (Fcgr), Hemopoietic Cell Kinase (Hck), Toll like receptors (Tlr), Linker For Activation of T Cells Family Member 2 (Lat2), Cd180 molecule (Cd180), Cell Surface Glycoprotein CD200 Receptor 1 (Cd200r1), C-C Motif Chemokine Receptor 5 (Ccr5), Triggering Receptor Expressed On Myeloid Cells 2 (Trem2), Colony Stimulating Factor 1 Receptor (Csf1r), Interleukin 1 Receptor Antagonist (Il1rn), Tumor Necrosis Factor Receptor Superfamily, Member 11a, NFKB Activator (Tnfrsf11a).

**Table 3 biomolecules-10-01292-t003:** Differentially expressed genes data of intracellular molecules downregulated by EPA in visceral adipose tissue of B6 obese mice. FDR-adjusted *p* value ≤ 0.05, fold change ≥ 1.5 at 95% confidence.

Gene Name	HF RPKM	HF-EPA RPKM	Fold Change	*p* Value
*Hck*	16.142	2.941	0.182	0.0438
*Ncf1*	31.825	7.866	0.247	0.0460
*Syk*	32.369	8.579	0.265	0.0385
*Pak1*	9.562	2.805	0.293	0.0435
*Casp1*	30.155	9.215	0.30	0.0385
*Card11*	13.84	1.578	0.11	0.0276
*Pik3r5*	17.611	4.782	0.27	0.0385
*Atf3*	63.616	10.225	0.160	0.0385
*Pik3ap1*	24.872	4.484	0.180	0.0293
*Blnk*	45.812	8.743	0.190	0.0248
*Naip2*	4.613	1.316	0.28	0.0385
*Pycard*	49.537	14.658	0.29	0.0385
*Irf5*	49.256	16.373	0.33	0.0421
*Alox5ap*	120.5	36.156	0.29	0.0473
*Ccl9*	405.632	54.25	0.13	0.0196

Abbreviations: Hemopoietic Cell Kinase (Hck), Neutrophil Cytosolic Factor 1 (Ncf1), Spleen Associated Tyrosine Kinase (Syk), P21 (RAC1) Activated Kinase 1 (Pak1), Caspase 1 (Casp1), Caspase Recruitment Domain Family Member 11 (Card11), Phosphoinositide-3-Kinase Adaptor Protein 1 (Pik3ap1), Phosphoinositide-3-Kinase Regulatory Subunit 5 (Pik3r5), Activating Transcription Factor 3 (Atf3), B Cell Linker (Blnk), Colony Stimulating Factor 1 Receptor (Csf1r), NLR family apoptosis inhibitory protein (Naip2), PYD and CARD Domain Containing (Pycard), Arachidonate 5-lipoxygenase activating protein (Alox5ap), C-C Motif Chemokine Ligand (Ccl9).

**Table 4 biomolecules-10-01292-t004:** Differentially expressed microRNAs modulated by EPA in visceral adipose tissue of B6 obese mice. FDR-adjusted *p* value ≤ 0.05, log2 fold change > 1.25.

miRNA	RPKM HF	RPKM HF-EPA	Fold Change	*p* Value
**mmu-mir-30a-5p**	11.26	11.85	0.59	0.0261
**mmu-mir-143-3p**	16.46	17.1	0.64	0.0041
**mmu-mir-125b-5p**	14.16	13.82	−0.33	0.0226
**mmu-mir-146b-5p**	11.42	10.65	−0.77	0.0173
**mmu-mir-181b-5p**	12.17	11.81	−0.36	0.0296
**mmu-mir-221-3p**	12.38	11.62	−0.76	0.0024

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
