# Peer review of "Eicosapentaenoic Acid Regulates Inflammatory Pathways through Modulation of Transcripts and miRNA in Adipose Tissue of Obese Mice"

_biomolecules, 2020, doi:10.3390/biom10091292_

Round 1
Reviewer 1 Report
In this work, the authors aimed to investigate global profiling of genes and miRNAs expression from VAT of mice fed HF diet and HF diet supplemented with EPA.They identified a vast number of genes differentially downregulated and microRNAs differentially expressed in VAT from HF-EPA compared to HF. Thus, through a novel approach this study adds data about the impact of EPA supplementation in HF diet fed mice. The MS is clearly written, well organized and scientifically sounds. Please take into account the following comments:
- The methodology is clearly described, however it would probably be interesting to know the amount of EPA in both diets and how it is reflected in human diet.
- To complete the puzzle of EPA role in inflammatory obesity it would be necessary to perform not only genomic, but also proteomic and functional assays. Indeed, one of the main weaknesses of the paper is mentioned by the authors in discussion, namely the lack of analysis of protein levels. Anyway, in regard to the functional studies, it would be interesting to have data about circulatory inflammation mediators such as cytokines and chemokines usually increased in obesity. Can the authors briefly elabor ate on this point in discussion? Which of them will be affected by EPA supplementation?
Specific minor comment:
Please report exact p values in all the Tables to three decimal places.
Author Response
Reviewer 1
In this work, the authors aimed to investigate global profiling of genes and miRNAs expression from VAT of mice fed HF diet and HF diet supplemented with EPA. They identified a vast number of genes differentially downregulated and microRNAs differentially expressed in VAT from HF-EPA compared to HF. Thus, through a novel approach this study adds data about the impact of EPA supplementation in HF diet fed mice. The MS is clearly written, well organized and scientifically sounds. Please take into account the following comments:
The methodology is clearly described, however it would probably be interesting to know the amount of EPA in both diets and how it is reflected in human diet.
Response: The high fat diet does not include any EPA but has other omega 3 fatty acids, to ensure it is not fatty acid deficient (Kalupahana, N.S. et al., 2010). EPA is supplemented at 36g/kg diet. We have provided details below about how we estimated and translated the dose of EPA from mouse to humans. Our calculation is based on energy intake (method #1); we also included other used methods under #2 (Allometric). Approximate human equivalent is ~10g EPA per day
- Our approach is energy/caloric-based, not BW-based:
- We fed mice 36 g EPA/kg diet that is at 45%kcal fat, which is equal to ~6.7% of energy for the mice.
- The human diet typically is estimated at 35% of total fat for a western diet with a caloric intake of 2000 calories. Thus, considering 35% kcal fat in human diet, the percentage of energy from EPA would be ~5% (as opposed to 6.7% for mice at 45% kcal fat)
- Thus, total calories from EPA based on 2000 Kcal: 2000*5/100=100 Kcal~ 10g EPA
- Other approach based on animal and human weight references.
Here, a dose by factor method applies an exponent for body surface area (0.67), which accounts for difference in metabolic rate, to convert doses between mice and humans. Thus, human equivalent dose (HED) is determined by an equation provided in references below [1,2]:
- EPA: 36g/kg of diet
- Mice eat ~ 2.5g/day, which would = 90 mg of EPA per mouse per day
- Mice BW at start was 30 g; thus in mg per kg BW it would be 90mg/30g or 3000mg/kg BW
- If I use the formula in references (Nair and Jacob, 2016; Nair, Morsy and Jacob, 2018), where HED is Human Equivalent Dose:
HED mg/kg = animal dose mg/kg * (animal weight(kg)/ human weight (kg))^0.33 (km value of human – mice)
then HED (mg/kg) = 3000* (0.030/60)^0.33= 3000 * 0.07= 244 mg EPA /kg BW
This would be for a 60kg person: 14g and for a 75kg person: 18g
Dose obtained using this approach is about double (1.8-fold higher) the amount we get using our kcal energy calculations in #1 above.
- Another method takes into account body surface area, which assigns different Km factors for humans and animals (Reagan-Shaw, Nihal and Ahmad, 2008).
References:
Kalupahana, N. S., Claycombe, K., Newman, S. J., Stewart, T., Siriwardhana, N., Matthan, N., Lichtenstein, A. H., Moustaid-Moussa, N. (2010) Eicosapentaenoic acid prevents and reverses insulin resistance in high-fat diet-induced obese mice via modulation of adipose tissue inflammation. J Nutr 140, 1915-22.
Nair, A.B.; Jacob, S. A simple practice guide for dose conversion between animals and human. J Basic Clin Pharm 2016, 7, 27-31, doi:10.4103/0976-0105.177703.
Nair, A.; Morsy, M.A.; Jacob, S. Dose translation between laboratory animals and human in preclinical and clinical phases of drug development. Drug Dev Res 2018, 10.1002/ddr.21461, doi:10.1002/ddr.21461.
Reagan-Shaw, S.; Nihal, M.; Ahmad, N. Dose translation from animal to human studies revisited. Faseb j 2008, 22, 659-661, doi:10.1096/fj.07-9574LSF.
Based on the description above, we changed the text of item 2.1 (line 64 to 66) of the manuscript, as follows:
Male C57BL/6 mice aged 5-6 weeks were fed a HF (45% kcal from fat, 35% kcal from carbohydrate, 20% kcal from protein) or a HF diet supplemented with 36g EPA/kg diet, (6.57% kcal) for 11 weeks (HF-EPA). Various methods are used to translate mouse dietary intakes into human equivalent intakes, as we detailed in the discussion section.
To complete the puzzle of EPA role in inflammatory obesity it would be necessary to perform not only genomic, but also proteomic and functional assays. Indeed, one of the main weaknesses of the paper is mentioned by the authors in discussion, namely the lack of analysis of protein levels. Anyway, in regard to the functional studies, it would be interesting to have data about circulatory inflammation mediators such as cytokines and chemokines usually increased in obesity. Can the authors briefly elaborate on this point in discussion? Which of them will be affected by EPA supplementation?
Response: This is an important point for our discussion. We appreciate such comment. The following statement was added to Discussion (item 4, page 11 to 12, line 383 - 410):
“Based on previous reports and on the main EPA targets found in this study, several cytokines and chemokines are affected systemically by EPA. Different studies have reported elevated systemic levels of IL-1RA that can be released by the adipose tissue in individuals with obesity [36-39]. Since our transcriptome data showed EPA reducing Il1rn expression in VAT of HF mice consistent with previous reports [40], we speculate that EPA would reduce IL-1RA levels in the circulation of HF mice. Thus, it is likely that IL-18 is similarly changed. First, IL-18 is usually found in the circulation of individuals with obesity [41]. Second, we report that EPA reduced inflammasome markers in VAT, which would reduce IL-18 release from adipose tissue. Indeed, reduced systemic levels of IL-18 when EPA is added to the diet of subjects with obesity has been reported [40]. TNF-α is another cytokine whose elevated levels are downregulated by EPA [40]. Our transcriptome and miRNA data have shown EPA affecting NFkB signaling, which drives TNFα transcription [32]. Therefore, TNFα systemic levels would also be reduced by EPA indirectly through NFkB regulation in our obesity model. The same could be suggested for MCP-1, since our lab and others have previous reports of EPA reduced MCP-1 along with macrophage content in adipose tissue and serum of diet-induced obese mice [5, 42]. CCL5 is another chemokine suggested as a target for EPA in our model. CCL5 transcription is also induced by NFkB. In agreement with this, EPA reduced the expression of CCL5 receptor, Ccr5, and other markers of NFkB pathways in our data.”
The reference numbers of the text correspond to the ones of the manuscript.
Specific minor comment:
Please report exact p values in all the Tables to three decimal places.
Response: The p values were modified in all tables as recommended. Edited tables were replaced in the manuscript and in the supplementary material file as well.
Reviewer 2 Report
In this work the authors carried out omic analyses to explore the beneficial effects of EPA treatment on VAT upon HFD. My main and serious concern is related to the setting of the RNAseq analyses that in my opinion are not methodologically correct thus invalidating the downstream analyses.
In particular:
If I understood well, the authors state that only 3 mRNA transcripts are in common between HF and HF-EPA mouse groups. Based on this assumption, how can it be possible that 153 genes are significantly differentially expressed? I cannot understand the conclusions of the authors about this result. The same is for miRNA omic analyses. Hence, I'm not able to understand overall the downstream functional enrichment analyses as well as the validation analyses carried out by the authors. Otherwise, the authors have to better explain how the differential expression analyses have been carried out.
Author Response
Reviewer 2
In this work the authors carried out omic analyses to explore the beneficial effects of EPA treatment on VAT upon HFD. My main and serious concern is related to the setting of the RNAseq analyses that in my opinion are not methodologically correct thus invalidating the downstream analyses.
In particular:
If I understood well, the authors state that only 3 mRNA transcripts are in common between HF and HF-EPA mouse groups. Based on this assumption, how can it be possible that 153 genes are significantly differentially expressed? I cannot understand the conclusions of the authors about this result. The same is for miRNA omic analyses. Hence, I'm not able to understand overall the downstream functional enrichment analyses as well as the validation analyses carried out by the authors. Otherwise, the authors have to better explain how the differential expression analyses have been carried out.
Response: We understand the concern raised by the reviewer. We have discussed about the values acquired from Venn Diagram analysis, and we decided to remove such analysis of our study to avoid confusion with future readers. Based on this decision, we removed the parts where Venn diagram was referred, and adjusted methods (page 3, item 2.5), description of results (page 4, item3.2), and Figure 1 and its legend (page 5).
Regarding the second part of the comment: after we acquired the RPKM values, fold change, and p value from a double-blind analyzer, we used Excel filter tool to identify differentially expressed genes among the global transcriptome. We used the filters “show numbers bigger than 0.05”, “show numbers bigger than 0.3” and “show numbers bigger than 1.5 or show numbers smaller than 0.6” to identify differentially expressed genes, confident expression, and minimum fold change, respectively. For miRNAs, the filters were “show numbers bigger than 0.05”, “show numbers bigger than 3.34” and “show numbers bigger than 1.25” to identify differentially expressed miRNAs, confident expression, and minimum fold change, respectively. (this part was added to item 2.5 of Material and Methods section).
About the third part of the comment: we used the IPA software to identify nucleotide sequence- and literature-based networks among transcripts. IPA was also used to identify nucleotide sequence- and literature-based networks between miRNAs and mRNAs (this part was added to item 2.9 of Material and Methods section). Alox5 appeared in the networks shown in IPA, and LTB4-related genes (Myd88, Stat1 and NFkB) also appeared as intermediates linking the pathways based on gene and miRNAs that we found. Based on this, we built the figure 4 network, and evaluated LTB4 levels in VAT, gene expression (qPCR) of the main intermediaries and key transcripts that compose such network (Figure 2 and 3) (this part was added to the last paragraph of item 3.5 of Results section). We hope this explanation is satisfactory.
Reviewer 3 Report
The authors invastigated global profiling of genes and miRNAs expression to explore the regulatory effects of eicosapentaenoic acid (EPA) in visceral adipose tissue (VAT) of obese mice feeding with different diet. Interestingly, they identified 153 genes differentially downregulated and 64 microRNAs differentially expressed in VAT from HF-EPA compared to HF mice.
Thera are some minor limitation of the study.
- Conclusion in the abstract and main body of manuscript should be modified. Authors did not explore if EPA reduce obesity and related metabolic disease. They examine only some pathway, moreover in the mice.
- In the first mention abbreviation should be explain for example in the abstract or IL-6, HOMA-IR, FFA or in the Tables.
- First line 283 adipose tissuse should be change for VAT
- It is not clear why in the Result section are cited previous data, it shoul be moved to the Disscusion section.
I have also two question:
- Why mice was feeding for 11 weeks?
- It is not clear how three first authors contributed equally in their work, and additioanlly last author is corresponding author. In the paragraph Authors Contribution (line 368-370) the above is not clear.
Author Response
Reviewer 3
The authors investigated global profiling of genes and miRNAs expression to explore the regulatory effects of eicosapentaenoic acid (EPA) in visceral adipose tissue (VAT) of obese mice feeding with different diet. Interestingly, they identified 153 genes differentially downregulated and 64 microRNAs differentially expressed in VAT from HF-EPA compared to HF mice.
Thera are some minor limitation of the study.
Conclusion in the abstract and main body of manuscript should be modified. Authors did not explore if EPA reduce obesity and related metabolic disease. They examine only some pathway, moreover in the mice
Response: We thank the reviewer for the important point raised. The conclusion in the abstract (page 1) was changed, as follows: “Together, our results identified key VAT inflammatory targets and pathways, which are regulated by EPA. These targets merit further investigation to better understand protective mechanisms of EPA in obesity-associated inflammation.”. The conclusion of the manuscript was also changed (item 5, page 11 and 12): “This study addressed EPA regulatory effects on several previously known genes along with few new genes, miRNAs and pathways involved in inflammation, which lead to dysregulated metabolic homeostasis. We used a novel approach for examining the impact of EPA supplementation in HF diet fed mice, by performing both transcriptomic and miRNA profiling. In addition to analysis of differentially expressed genes and miRNAs, we conducted pathway analysis that provides a comprehensive understanding of metabolic and molecular changes mediated by EPA. Specifically, we identified inflammation-related targets in VAT that pave the way for further mechanistic studies for potential use of EPA in obesity associated-inflammation and metabolic dysfunctions.”
In the first mention abbreviation should be explain for example in the abstract or IL-6, HOMA-IR, FFA or in the Tables.
Response: “interleukin (IL)” was first mentioned in the introduction (line 40, page 1) when referred to IL-10. HOMA-IR and FFA were wrote out before abbreviation (Item 4 Discussion, page 10): homeostatic model assessment for insulin resistance score (HOMA-IR) (line 325); free fatty acids (FFA) (line 353). The gene symbols that appear in the manuscript were extended when first mentioned in the text and in the tables of the manuscript. However, we could not extend the symbols that appear in the abstract due to the limited 200 words required by the journal.
First line 283 adipose tissue should be change for VAT
Response: We changed the text (item 4 Discussion, page 10), as follows: “This study reports the impact of EPA supplementation in HF diet fed mice by gene and miRNA profiling analysis in VAT.”
It is not clear why in the Result section are cited previous data, it should be moved to the Discussion section.
Response: We have removed previous results from the Results section and moved to discussion section (page 10, line 323 to 326). The text added to Discussion section was modified, as follows: “We have previously reported significant reductions in body weight, basal glycemia, insulinemia, insulin resistance (evaluated by homeostatic model assessment for insulin resistance score - HOMA-IR), glucose clearance, and body fat in mice fed HF-EPA compared to HF [5, 6].”.
The reference numbers of the text above correspond to the ones of the manuscript.
I have also two question:
Why mice was feeding for 11 weeks?
Response: 8-12 weeks is a common intervention period for diet-induced obesity, and some studies extend to 16-20 weeks. We have successfully used 11-12 weeks for diet-induced obesity studies in mice using 45% kcal fat diets and mice develop insulin resistance and glucose tolerance and exhibit inflammation, body and fat gain during this time. Thus, we used the 11-week timeframe which is consistent with diet-induced obesity studies that provide sustained metabolic dysfunctions such as adipose tissue hypertrophy. Also this time allows for EPA to be incorporated in cell membranes according to previous works of our group (Kalupahana, N.S. et al., 2010).
Reference #5 of the manuscript: Kalupahana, N. S., Claycombe, K., Newman, S. J., Stewart, T., Siriwardhana, N., Matthan, N., Lichtenstein, A. H., Moustaid-Moussa, N. (2010) Eicosapentaenoic acid prevents and reverses insulin resistance in high-fat diet-induced obese mice via modulation of adipose tissue inflammation. J Nutr 140, 1915-22.
It is not clear how three first authors contributed equally in their work, and additioanlly last author is corresponding author. In the paragraph Authors Contribution (line 368-370) the above is not clear.
Response: Actually, there are only 2 co-first authors, Theresa Ramalho and Mandana Pahlavani, who contributed equally as first authors, identified by “*”. Because this was a collaboration between 2 labs, there are two co-corresponding authors, Dr. Sonia Jancar and Dr. Naima Moustaid-Moussa, the latter is the contact corresponding author.
The text of Author Contributions Item (page 12, line 433 to 437) was modified, as follows:
“NSK and NMM designed and conducted the animal study; NMM and SJ designed the research reported in this paper and the outline of the article, and both guided the writing process. MP and NW conducted RNA Seq. TR and MP analyzed the sequencing and pathway analyses of the data. TR, LR, MP conducted the functional analyses and validation of genes, miRNAs and LTB4 assays. TR and MP wrote the first draft of the paper. All authors edited, proofread, and approved the manuscript before submission.”
Reviewer 4 Report
Although the results in this paper are not very striking, it follows the standard analysis flow of mRNA and miRNA and the manuscript was well written. There are no reasons to reject it. I recommend its publication as it is.
They tried to find mRNA and miRNA interactions in EPA treatments and found some interaction. Although nothing is so interesting, this is the first experiment report as long as I know. In my knowledge, this is the first report on mRNA-miRNA interaction in EPA treatment. Simply, it is the first experiment of this kind. The conclusions are consistent with the evidence and arguments presented. No question was posted in advance, since it is data driven study.
Author Response
Reviewer 4:
Although the results in this paper are not very striking, it follows the standard analysis flow of mRNA and miRNA and the manuscript was well written. There are no reasons to reject it. I recommend its publication as it is.
They tried to find mRNA and miRNA interactions in EPA treatments and found some interaction. Although nothing is so interesting, this is the first experiment report as long as I know. In my knowledge, this is the first report on mRNA-miRNA interaction in EPA treatment. Simply, it is the first experiment of this kind. The conclusions are consistent with the evidence and arguments presented. No question was posted in advance, since it is data driven study.
Response: Thank you very much for your positive comments.
Round 2
Reviewer 2 Report
The authors were not able to explain the meaning of the previously presented Venn diagrams. I would like to have an explanation.
Now they have presented the results by explaining more in details the methods used. I think that it would be better to use the FDR value and not the p value to select the genes and miRNAs that are differentially expressed. The FDR value is highly recommended when the omic analyses identify a very high number of genes (e.g. about 13000 genes for RNAseq).
Author Response
We appreciate the reviewer’s constructive comment. We addressed the remaining concerns and edited the manuscript text accordingly. Relevant modified text in the manuscript is highlighted in red.
Responses to Reviewer 2 highlighted in red:
- The authors were not able to explain the meaning of the previously presented Venn diagrams. I would like to have an explanation.
Response: Fair question, thank you. In the previous revision, we decided to remove the Venn diagram, because we agreed with the reviewer that it was confusing as presented. Also, it was not accurate because as it showed only 3 genes that are common to HF and HF-EPA, which is not correct; there were 153 genes as differentially regulated, as discussed in the paper. Accordingly, the Venn diagram was not consistent with the text description (latter was accurate), and it was also confusing because the Venn and figure were previously made using different cut offs and number of DE genes than what we described in the text. In the latter, we were describing the whole analysis without cut off or with>0.3 RPKM cut off. So again, apologies for the confusion. We just wanted to avoid readers’ confusion and for better clarity, it is better to match the figure with description in the text, and thus, we removed the Venn diagram (which does not add much) and focused on analyses that are presented in the paper. We hope this is a satisfactory response.
Moreover, we reviewed again all the analyses and all is accurate and we confirmed the fold change as well. We did clarify in the figure that there were 153 DE genes, which were all downregulated with the cut offs indicated. However, we could identify a few upregulated genes as discussed in the paper, and as presented as well, which had low RPKM.
- Now they have presented the results by explaining more in details the methods used. I think that it would be better to use the FDR value and not the p value to select the genes and miRNAs that are differentially expressed. The FDR value is highly recommended when the omic analyses identify a very high number of genes (e.g. about 13000 genes for RNAseq).
Response: We agree, FDR is the recommended statistical way for large scale gene expression analyses and the p values in our paper indeed reflect the FDR-adjusted p values. This was explained in the method. We did indicate in the previous version that analyses included FDR adjustment (highlighted now in red). Basically, the p values in our paper are FDR-adjusted values. However, it may not have been clearly described. Here is what we did: Following the sequencing and data normalization and analyses, we used false discovery rate (FDR) with cutoff 5% (Benjamini & Hochberg test) and QSeq software (DNASTAR) to identify differentially expressed (DE) genes between HF (control) and HF-EPA at 95% confidence (Sgro, 2014). Therefore, since it has already been FDR- adjusted, the outputs did not have FDR values. We added the following statement to the legend of Figure 1: “FDR-adjusted p value ≤ 0.05”. The following statement was added to the legend of table 1: “–log p value > 1.3 is equivalent to FDR-adjusted p value ≤ 0.05, Z score ≤ -2 indicates pathway inhibited, Ratio is number of differentially expressed/total number of genes in a canonical pathway.” We also added “FDR-adjusted p value ≤ 0.05, fold change ≥ 1.5 at 95% confidence.” to the legends of table 2, table 3, supplementary table 1 and 2, and added “FDR-adjusted p value ≤ 0.05, log2 fold change > 1.25.” to the legend of table 4 and supplementary tables 3 and 4.
Reference https://static-bcrf.biochem.wisc.edu/tutorials/dnastar/public_arrays/dnastar-arraystar-01_2014.pdf
